# Genome-Wide Association Study on Reproduction-Related Body-Shape Traits of Chinese Holstein Cows

**DOI:** 10.3390/ani11071927

**Published:** 2021-06-28

**Authors:** Xubin Lu, Ismail Mohamed Abdalla, Mudasir Nazar, Yongliang Fan, Zhipeng Zhang, Xinyue Wu, Tianle Xu, Zhangping Yang

**Affiliations:** 1College of Animal Science and Technology, Yangzhou University, Yangzhou 225002, China; dx120180094@yzu.edu.cn (X.L.); ismailhmk@gmail.com (I.M.A.); drmudasirnazar457@gmail.com (M.N.); dx120170088@yzu.edu.cn (Y.F.); dx120190127@yzu.edu.cn (Z.Z.); wzydl386@163.com (X.W.); 2Joint International Research Laboratory of Agriculture and Agri-Product Safety, Yangzhou University, Yangzhou 225009, China; tl-xu@outlook.com

**Keywords:** reproduction, body-shape traits, GWAS, SNP, Chinese Holstein, cows

## Abstract

**Simple Summary:**

Reproduction plays a pivotal role in dairy cow farming. Good reproductive performance in cows can decrease the elimination rate of cows, increase the success rate of breeding, and thereby enhance milk production. Identification of the genetic variants in reproduction-related traits helps to increase the genetic improvement of cows’ reproductive performance. In this study, we estimated the genetic parameters of three indicators of reproductive ability, namely, Loin Strength (LS), Rump Angle (RA), and Pin Width (PW), and conducted a genome-wide association study of them. The heritability of these three traits was medium, and in total, 11 significant single-nucleotide polymorphisms (SNPs) were detected. Through a bioinformatics analysis of the genes adjacent to these variants, 16 candidate genes were identified as being associated with these three traits. We expect that the results could help with the genetic improvement of Chinese Holstein cows’ reproductive performance.

**Abstract:**

Reproduction is an important production activity for dairy cows, and their reproductive performance can directly affect the level of farmers’ income. To better understand the genomic regions and biological pathways of reproduction-related traits of dairy cows, in the present study, three body shape traits—Loin Strength (LS), Rump Angle (RA), and Pin Width (PW)—were selected as indicators of the reproductive ability of cows, and we conducted genome-wide association analyses on them. The heritability of these three traits was medium, ranging from 0.20 to 0.38. A total of 11 significant single-nucleotide polymorphisms (SNPs) were detected associated with these three traits. Bioinformatics analysis was performed on genes close to the significant SNPs (within 200 Kb) of LS, RA, and PW, and we found that these genes were totally enriched in 20 gene ontology terms and six KEGG signaling pathways. Finally, the five genes *CDH12*, *TARP*, *PCDH9*, *DTHD1*, and *ARAP2* were selected as candidate genes that might affect LS. The six genes *LOC781835*, *FSTL4*, *ATG4C*, *SH3BP4*, *DMP1*, and *DSPP* were selected as candidate genes that might affect RA. The five genes *USP6NL*, *CNTN3*, *LOC101907665*, *UPF2*, and *ECHDC3* were selected as candidate genes that might affect the PW of Chinese Holstein cows. Our results could provide useful biological information for the improvement of body shape traits and contribute to the genomic selection of Chinese Holstein cows.

## 1. Introduction

Reproductive performance is an economically important feature of livestock. Successful pregnancy and birth of offspring can maintain the scale of a farm, increase the productivity and profitability of animal production, and improve farmers’ income [1]. Many reproductive traits of cattle, such as conception rate, reproduction rate, and ease of calving, are quantitative traits regulated by multiple genes and affected by environmental factors [2,3]. However, the records of reproductive performance are generally subjective, and due to the limited group size and poor management conditions of many small and medium-sized farms, are usually incomplete and inaccurate. This makes it difficult to conduct research on the reproductive performance of dairy cows and improve these traits effectively [4].

Body-shape linear scoring is an indispensable activity in dairy cattle breeding that is generally carried out as part of the national breeding program, and the level of linear scores expresses the specificity of dairy cow muscle and bone development and function [5]. The current linear scoring standard for Chinese dairy cows refers to the Code of practice of type classification in Chinese Holstein (GB/T 35568-2017) [6], in which a nine-point scoring system is used to evaluate 20 cow linear type traits. Of these, three traits are closely related to the reproductive performance of dairy cows; namely, Loin Strength (LS), Rump Angle (RA), and Pin Width (PW) [7,8,9,10]. LS is used to identify the firmness of the cow’s loin. Cows with a weak loin often have a sinking uterus, and the secretions in the uterus are difficult to discharge, which could easily cause reproductive system diseases and ultimately affect the ease of calving and conception rate of breeding cows [7,8,9]. In the previous studies of Holstein cows, LS had a high genetic correlation (0.43) with days open of cows [7], and the genetic correlations of LS with the first service period and calving ease of cows were 0.17 [8] and −0.11 [9], respectively. The RA score reflects the inclination angle of the cow’s loin to the end of the ischial tuberosity of the hip. A proper rump angle is conducive to the discharge of secretions and postpartum lochia in the cow’s reproductive tract, thereby increasing the reproductive rate of cows [8,9,10]. The genetic correlations of RA with the calving ease and dry period of cows were −0.28 [9] and 0.19 [8], respectively. The cows with the score of RA between 4.95–5.02 could have a significantly easier course of parturition [10]. The PW score reflects the width of the two ischial tuberosities at the buttocks of a cow. PW is related to the reproduction of cows [7,8]; studies have presented that the genetic correlations of PW with the first service period, dry period, and calving ease were 0.28, 0.26, and 0.15, respectively [7,8]. Therefore, the accurate determination and analysis of LS, RA, and PW could reflect the reproductive performance of dairy cows.

A genome-wide association study (GWAS) is a powerful method to screen a whole genome for genetic factors related to phenotypic traits by using single-nucleotide polymorphisms (SNPs) as genetic markers, and has been widely applied in domestic animals. Many GWAS studies have been carried out on dairy cows in recent years, but previous studies mainly focused on the important economic and disease traits of cows, including milk production, milk protein, body height, body weight, and ketosis [11,12,13,14]. In beef cattle, some QTLs and candidate genes have been predicted to be associated with reproductive performance, such as *MHC class II* genes, which were significantly associated with pregnancy success in Nellore cattle [15]. The 44 to 50 Mb region on the fifth chromosome was screened and found to be associated with the age at puberty of Nellore–Angus crossbred cattle [16], and *LOC511981*, *KIF1A*, and *EPRS* genes were related to the age at first calving of Xinjiang Brown cattle [17]. Although there are reports about GWAS research on pregnancy rate and calving interval of dairy cows in Iran and Europe [18,19], there are few GWAS reports on the reproductive performance of Chinese Holstein cows.

In this study, we conducted GWAS studies on LS, RA, and PW traits to identify the significant SNPs and candidate genes related to these traits of Chinese Holstein cows. We expect our results to become valuable resources for genetic evaluation and provide a theoretical basis for improving the genomic selection of reproductive performance in dairy cows.

## 2. Materials and Methods

### 2.1. Ethics Statement

The collection of hair-follicle samples and the measurement of traits in this study were conducted in accordance with the Institutional Animal Care and Use Committee of the School of the Yangzhou University Animal Experiments Ethics Committee (License Number: SYXK (Su) IACUC 2012-0029), and no animals were anesthetized or euthanized during the study.

### 2.2. Animals and Phenotypic Data

A total of 1730 healthy Chinese Holstein cows from four dairy farms in Jiangsu Province, China were used in this study (Farm 1: 407 cows; Farm 2: 209 cows; Farm 3: 739 cows; Farm 4: 375 cows). Three body-shape traits, Loin Strength (LS), Rump Angle (RA), and Pin Width (PW), of 1730 cows were measured according to the China National Standard (GB/T 35568-2017); at least three professionals performed the measurement of traits for each cow, and the average of the measurements taken by the different technicians was used as the phenotype of each trait to ensure the accuracy of the data. All cows were in the dry period when they were measured. The parities of cows were between 1 and 4, and the pedigree of the cows could be traced back at least three generations. The phenotype distribution of the three traits across the farms is shown in Appendix A. Of the 1730 cows, 214 cows in the Farm 4 were selected. They were all in their first lactation at the time of measurement, and the reproductive and calving traits’ records in the first parity of these cows, including the occurrence of pregnancy after one breeding, the occurrence of premature birth, and the ease of calving, were collected from farm to test their relationship with LS, RA, and PW (Appendix A).

### 2.3. Adjustment of Phenotypes for Analysis

The three body-type traits used for subsequent analysis were all adjusted with fixed environmental factors using the following two steps:

Step 1: We estimated genetic parameters in the following multiple-trait animal model:
(1)y=Xb+Za+e
where y is a vector containing individual phenotype observations of the three traits; X is a design matrix for the fixed effects (farm, age, and parity); b is the vector of fixed effects; Z is a matrix designed to link a to y and the variation between animals determined by the pedigree; a is the vector of individual additive genetic effects; and e is a vector of random residuals. It was assumed that the parameters in the model had the following independent normal distributions: $a\;\sim \;N\;\left(0,A\sigma_{a}^{2}\right)$ and $e\;\sim \;N\;\left(0,I\sigma_{e}^{2}\right)$, where **I** is a matrix for unit vector, A is the relationship matrix built through the pedigree,$\sigma_{a}^{2}$ is additive genetic variance, and $\sigma_{e}^{2}$ is residual variance. The variance components of $\sigma_{a}^{2}$ and $\sigma_{e}^{2}$ were estimated by the restricted maximum likelihood (REML) procedure in DMU software (v5.6) [20]. Finally, the impact of fixed environmental factors could be avoided. The heritability of each trait was calculated as $h^{2}=$
$\sigma_{a}^{2}/\left(\sigma_{a}^{2}+\sigma_{e}^{2}\right)$. The genetic correlation of each two traits was calculated as $r_{A}=Cov\left(a_{1},a_{2}\right)/\sqrt{\sigma_{a_{1}}^{2}*\sigma_{a_{2}}^{2}}$, where $Cov\left(a_{1},a_{2}\right)$ is the additive effect covariance of every two traits, and $\sigma_{a_{1}}^{2}$ and $\sigma_{a_{2}}^{2}$ are the additive genetic variance of traits 1 and 2, respectively. The standard errors of heritability and genetic correlations were computed based on a Taylor series approximation [20].

Step 2: We adjusted the phenotypes using the following model:
(2)$$y_{adj}={\left(Z^{\prime }Z\right)}^{-1}Z^{\prime }\left(y-X\hat{b}\right)$$
where $y_{adj}$ is the vector of adjusted phenotypes of traits; y, Z, and X are the same as in Formula (1), and $\hat{b}$ is an estimate of b, which was calculated in Formula (1).

### 2.4. Genotypic Data

Of the 1730 cows measured for body-type traits, hair-follicle samples of 999 cows were collected (Farm 1: 198 cows; Farm 2: 214 cows; Farm 3: 224 cows; Farm 4: 363 cows). DNA was extracted and genotyped using the GGP Bovine 100K SNP Chip by Neogen Biotechnology (Shanghai, China) Co., Ltd. (http://www.neogenchina.com.cn, accessed on 20 March 2021), and the ARS-UCD1.2 (bosTau9) was used as the reference genome. Then, the following quality-control criteria were implemented for the variants detected and individuals by Plink software (v 1.90) [21]: (1) the call rate of a single variant had to exceed 90%; (2) the minor allele frequency (MAF) of every SNP genotype had to exceed 5% and meet the Hardy–Weinberg Equilibrium (HWE) (*p* > 1.0 × 10^−6^); (3) SNP information on sex chromosomes had to be eliminated; and (4) the call rate of individual genotypes had to exceed 95%. Then, the variants and individuals that did not meet the quality-control requirements were removed. In this study, the SNPs on the sex chromosomes were also removed for the following three reasons: (1) the inheritance pattern of sex chromosomes is more complicated than that of autosomes [22,23]; (2) one of the two X chromosomes in female mammalian cells will lose activity (lyonization) for ‘dosage compensation’ of X-linked genes with males, which would cause a false positive of GWAS results [24]; and (3) the lyonization of X chromosomes is sometimes related to individual reproductive performance and disease occurrence, such as abortion and skin disease [25,26]. In total, 984 individuals and 84,407 variants were retained for subsequent analysis.

### 2.5. Linkage Disequilibrium Decay Analysis and Principal Component Analysis

Plink software (v1.90) [21] was used to detect the change of linkage disequilibrium (LD) with the increase of average distance between SNPs in the current population based on R^2^. Principal component analysis (PCA) was conducted using the FactoMineR package in the statistical software program R (v4.0.4) using the 84,407 variants of the 984 cows to estimate the population structure [27]. Then, the ggplot2 package in R (v4.0.4) was used for visual analysis of the results [28].

### 2.6. Genome-Wide Association Studies

The multilocus linear mixed model was used to conduct the association analysis between the SNPs and traits by the fixed- and random-model circulating probability unification (FarmCPU) method [29]. The FarmCPU method conducted GWAS by iterative usage of fixed- and random-effect models, and it could eliminate the confounding between testing markers and kinship. The fixed-effect model contained testing markers, one at a time, and multiple pseudo-quantitative trait nucleotides (QTNs) as covariates to control false positives. Possible association markers were calculated in each round of the fixed-effect model, and pseudo-QTNs were selected from the possible association markers in random-effect model by the SUPER (Settlement of MLM Under Progressively Exclusive Relationship) algorithm [30]. Pseudo-QTNs were used to define kinship of individuals to avoid a model over-fitting problem in the fixed-effect model [29]. To reduce the false positives caused by the population stratification, the three highest principal components (PCs) were used as covariate variables in the GWAS models. The following is the fixed-effect model [29]:(3)$$y_{adj}=Xb_{X}+M_{t}b_{t}+S_{j}d_{j}+e$$
where $y_{adj}$ is the vector of adjusted phenotypes of traits; X is a fixed-effects matrix constructed by the three highest PCs; $M_{t}$ is the matrix of t pseudo-quantitative trait nucleotide (QTN) genotypes, initiated as an empty set; $b_{X}$ and $b_{t}$ are the corresponding effects of the three PCs and t pseudo-QTNs, respectively; $S_{j}$ is the genotype of the j marker; $d_{j}$ is the effect of the j marker; and e is a vector of random residuals with a distribution with zero mean and variance of $\sigma_{e}^{2}$. The SUPER algorithm was used to update the selection of pseudo-QTNs in a random-effect model using the following three steps [30]: (1) We sort the SNPs by their *p*-Values calculated using Formula (3) for one trait. (2) For each bin (segment) on a chromosome, we chose one SNP with the lowest *p*-Value as the representative for the bin. Then, we selected the most influential bins to build kinship. The size and number of bins chosen were treated as parameters to maximize the restricted maximum likelihood for a trait. The selected SNPs (each representing a bin) were then used as a base of a SNP pool to define individual relationships for the later association test. (3) We excluded the SNPs in the SNP pool that were in LD (r^2^ > 0.7) with the testing SNP to derive a complementary trait-specific kinship. The random-effect model is as follows [29,30]:(4)y=u+e
where y and e are the same as in Formula (3); and $u\;\sim \;N\;\left(0,K\sigma_{u}^{2}\right)$, in which u is the genetic effect of the individual, K is the kinship matrix derived from the pseudo QTNs, and $\sigma_{u}^{2}$ is an unknown genetic variance.

The SNP genotypes coded for the association analyses were 0, 1, and 2, which were converted by Plink software (v 1.90) [21]. The explained genetic variation (EVG) of each SNP was calculated as follows:(5)$$EVG=\frac{2p\left(1-p\right)\;*\;effect^{2}}{\sigma_{a}^{2}}$$
where
p
is the minor allele frequency (MAF) of each variant;
effect
is the result of
$d_{j}$
for each significant variant in Formula (3), which means the regression coefficient of adjusted phenotype to each variation; and $\sigma_{a}^{2}$ is additive genetic variance.

We set the significance threshold for selecting the significant SNPs using the Bonferroni correction method [31]. The type I error rate was controlled at 5%, and the genome-wide significance threshold was 5.90 × 10^−7^ (0.05/84407).

### 2.7. Annotation of Candidate Gene and Bioinformatic Analysis

Genes within 200 Kb (LD > 0.35) upstream and downstream of the significant SNPs detected from the three indicators of reproductive ability (LS, RA, and PW) were selected as candidate genes at “https://www.ensembl.org/Bos_taurus/Info/Index” (accessed on 20 March 2021), and ARS-UCD1.2 (bosTau9) was used as the reference genome in this process. Then, the Gene Ontology (GO) and the Kyoto Encyclopedia of Genes and Genomes (KEGG) pathway analysis (https://www.genome.jp/kegg, accessed on 20 March 2021) was conducted on the candidate genes using the cluster profiler package in R (v4.0.4) [32].

## 3. Results

### 3.1. The Relationship between Phenotype Data and Reproductive Performance

The relationship between the phenotype traits studied in this article (LS, RA, and PW) and the reproductive and calving traits of cows in the first parity, including the occurrence of pregnancy after one breeding, the occurrence of premature birth, and the ease of calving, were evaluated using the method of Independent Sample *t*-Test in SPSS (v26.0) software (IBM, NewYork, NY, USA). Because these 214 cows were all from same farm (Farm 4), they were all in their first lactation at the time of measurement, the technicians were the same, and the measurement work was all finished in one week, we used the raw measured scores (not the adjusted phenotypes) to check the relationship between LS, RA, and PW and reproductive traits.

As shown in Figure 1, the RA and PW scores significantly affected the occurrence of pregnancy after one breeding (Figure 1a, *p* < 0.05), the LS and PW scores were significantly associated with the occurrence of premature birth in the parity (Figure 1b, *p* < 0.05), and the LS and PW scores were significantly related to the ease of calving of cows (Figure 1c, *p* < 0.05).

### 3.2. Phenotypic Data and Genetic Parameters Estimation

The adjusted body-type traits of the 984 cows including LS, RA, and PW presented
approximately normal distributions in this study (Figure 2). The descriptive statistics, as well as estimation of the genetic parameters of the traits, are shown in Table 1. The animal model was used to estimate the genetic parameters for each trait and the heritability estimated for LS, RA, and PW, and was 0.38, 0.22, and 0.20, respectively. It was also found that the phenotypic correlations of the three traits were −0.06 (LS and RA), −0.14 (LS and PW), and 0.13 (RA and PW), and the genetic correlations were 0.36 (LS and RA), −0.08 (LS and PW), and 0.27 (RA and PW) (Appendix A).

### 3.3. SNP Data Statistics

After quality control, 84,407 SNPs on 29 chromosomes remained for subsequent marker analysis. The distribution of the SNP information within 200 Kb windows on the different chromosomes is shown in Figure 3a. The change of LD decay with the increase of the average distance between SNPs in the current population is presented in Figure 3b; the R^2^ was lower than 0.35 when the average distance between SNPs was around 200 Kb.

### 3.4. Population Structure Analysis

The three highest principal components (PCs) were used to determine the population stratification level. As shown in Figure 4, the population in this study was stratified into several unevenly sized groups. Therefore, in order to avoid the false positive caused by group stratification, these three PCs were used as covariates in the fixed-effect model for association analysis. In total, the three highest PCs explained 28.3% of the variation, of which they respectively occupied 11.8%, 9.2%, and 7.3% (Figure 4).

### 3.5. Genome-Wide Association Study

To ensure the accuracy of the association analysis between phenotypes and variants, quantile–quantile (QQ) plots of the three traits were drawn according to the *p*-Value of each SNP. The vast majority of the variants did not deviate from the expected *p*-Value, which revealed that the models and methods for GWAS analysis were reasonable (Figure 5).

As mentioned before, the threshold for selecting significant SNPs in the GWAS study was 5.9 × 10^−7^ (0.05/84407). Four SNPs (rs43162548, rs133475777, rs109073659, and rs42946768) located on chromosome 4, 6, 12, and 20, respectively, were detected to be associated with trait LS, and the genes nearest to the four SNPs were *TARP* (5 Kb), *DTHD1* (within), *PCDH9* (within), and *CDH12* (within), respectively. Four SNPs (rs43352090, rs43366267, rs43486059, and rs13724035) located on chromosome 3, 3, 6, and 7, respectively, were detected to be associated with trait RA, and the genes nearest to the four SNPs were *ATG4C* (200 Kb), *SH3BP4* (50 Kb), *LOC781835* (within), and *FSTL4* (within), respectively. Three SNPs (rs109578471, rs43430205, and rs42051017) located on chromosome 12, 22, and 29, respectively, were detected to be associated with trait PW, and the genes nearest to the three SNPs were *USP6NL* (within), *CNTN3* (200 Kb), and *LOC101907665* (200 Kb), respectively (Table 2, Figure 6).

### 3.6. Enrichment Analysis

For an in-depth understanding of the function of the 11 significant SNPs related to the indicators of the reproductive ability of cows (LS, RA, and PW), the genes within 200 Kb of significant SNPs for each trait were selected for enrichment analysis, and a total of 45 genes were obtained, of which 11 belonged to LS, 23 to RA, and 11 to PW (Appendix A). These candidate genes of LS, RA, and PW were enriched into 12, 8, and 0 GO terms, respectively, and were clustered into 3, 2, and 0 categories using FunSet online software (http://funset.uno; Appendix A) [33]. The three categories of LS were: cell-adhesion progress, cell–cell adhesion via plasma-membrane adhesion molecule progress, and cell–cell junction organization progress, and the two categories of RA were the protein-deglycosylation process and protein-modification process (Figure 7). The KEGG results (Table 3) showed that the candidate genes of each trait were significantly enriched in the following six pathways (*p* < 0.05): endocytosis, other glycan degradation, ECM-receptor interaction, autophagy—other, mRNA surveillance pathway, and RNA transport; 7 of 45 candidate genes were involved in pathway regulation (Table 3).

## 4. Discussion

Reproduction is a key factor in dairy cows’ postpartum lactation and herd expansion. Cows with good reproductive performance show pregnancy symptoms on time and can become pregnant after the first mating, which can directly increase production performance and the economic situation of dairy farms [34]. Due to the limited management in some small and medium-sized farms in China, records of the reproductive traits, such as conception rate, reproduction rate, and ease of calving, are often incomplete and subjective. To improve the reproductive performance of dairy cows, in this study, we selected three body-type traits that were easy to measure; namely, LS, RA, and PW, as indicators of the reproductive ability of dairy cows, and conducted genome-wide association analyses of them, hoping to find new QTLs that might affect these traits.

Studies have reported that body-type traits could be the indicator traits of reproduction of pigs and cattle. The vulva score categories (VSC) had the potential to improve reproductive efficiency in the first parity performance, and had been proposed as an indicator trait of efficient reproductive performance in sows [35]. The estimated genetic correlation of number born alive (NBA) was 0.47 with front width and 0.55 with chest width, implying that front width and chest width could be promising indicator traits for efficiently improving NBA [36]. The loin depth had strong positive genetic correlations with litter weight gain (LWG) (0.24 to 0.54), and it could be used as indicator traits of reproduction in sows [37]. The score of subcutaneous body fat thickness of the dairy cow has a medium genetic correlation with the interval between first and second calving (−0.27) and conception rate (0.22), and it could affect the future reproductive performance of cows [38,39], and the body condition score of the subcutaneous body-fat thickness in the first month after calving could be the favorable indicators of cows’ reproduction [38,40]. In this study, we selected three body-type traits that were easy to measure; namely, LS, RA, and PW, as indicators of the reproductive ability of dairy cows. Although we did not estimate the impact of the three traits on reproductive traits at the genetic level, there was a clear impact of them on reproductive traits at the phenotypic level (Figure 1), such as RA and PW being significantly related to the occurrence of pregnancy after one breeding. A higher LS score could significantly reduce the incidence rate of premature birth in dairy cows, and cows with a higher LS score were easier to calve (Figure 1). This shows that the three traits used in this study were reasonable indicators of dairy cow reproductive performance. As in the previous study results [10], we also found that a trait score that is too high might be detrimental to the cows, such as a PW score >7.5, which might result in lower pregnancy rates, premature delivery, and difficulty of calving (Figure 1). Therefore, genetic improvement and genome-selection work on the body traits of dairy cows must be combined with the actual production situation of the farm to find the best score for each trait, and should not blindly pursue the high trait scores.

However, some studies also presented the limitations of using body-type traits as indicator traits to improve reproductive performance, such as the environmental factors that would influence the score of measurement [41,42], although a well-trained scorer would have problems in consistently detecting scores within a deviation of about 0.25 [39]. We also found that in this study, the total genetic variation explained by the detected SNPs of LS, RA, and PW was only 4.40%, 4.68%, and 2.70%, respectively (Table 2), and these SNPs might mainly affect these three body-type traits, so the effect on the reproduction of dairy cows might be less, and it needs to be confirmed by follow-up experiments. Therefore, it is more accurate and suitable to directly conduct research on the reproduction traits to improve reproductive performance if the records of reproduction in farms are complete.

In this study, the heritability of reproductive traits of cattle was medium (Table 1). The heritability of the calving ease of Nellore cattle ranged from 0.18 to 0.39 [43]; Yamazaki et al. reported that the heritability of the conception rate at first insemination of Japanese Holstein cows was 0.393 [44]; and Pablo estimated that the heritability of the calving interval of Japanese Black cows ranged from 0.12 to 0.20 [45]. The heritability of the three reproduction-related traits studied in this paper was between 0.20 and 0.38 (LS 0.38, RA 0.22, and PW 0.20; Table 1). Therefore, to improve the selection and breeding of these three traits of dairy cows, it is necessary to pay attention to the influence of environmental factors, such as age, climate, and nutritional status, on the measurements [1]. Although phenotypic correlations existed among LS, RA, and PW, the genetic correlations were low (<0.2; Appendix A). Therefore, these three traits should be selected separately in dairy cattle.

The linkage disequilibrium (LD) analysis of the population is the basis of association studies. In the present study, the level of LD (r^2^) between SNPs decreased as the distance increased, and the R^2^ was lower than 0.35 when the average distance between SNPs was around 200 Kb (Figure 3b). Therefore, 200 Kb was used to search for candidate genes that were in LD with the significant SNPs, and this was also a common distance used to search for genes in other GWAS studies [46,47]. The decay rate of Chinese Holstein cows in this study was much lower than in Simmental cattle, Wagyu cattle, and Iranian water buffalo, which indicated that the degree of artificial selection of Chinese Holstein cows was higher than in beef cattle [48]. From the SNPs’ density distribution on 29 autosomes, we could find that the SNP information of the GGP Bovine 100K SNP Chip used in this study was evenly distributed across each chromosome, and the number of SNPs within 200 Kb was generally less than 22. There were still some blank areas that had no variant information on certain chromosomes, such as Chr 7, Chr 10, Chr 12, and Chr 16 (Figure 3a), and these could be used as key areas to discover new variants in the future.

Population stratification is an important confounding factor in GWAS studies. When samples with different genetic structures are included in a GWAS study, the genetic differences caused by the evolutionary selection of individuals from different groups and regions might be interpreted as phenotypic differences in the GWAS process and result in false-positive association results [49]. In the PCA scatter plot, the population was separated into several different subgroups, which showed that there was population stratification in this study (Figure 4). The stratification may have been caused by the semen used on the four farms coming from different countries, and some of the semen may have been from local bulls. To correct the effects of population stratification, the genetic population structure of each individual was fitted as a fixed effect in the GWAS models. The inflation factors (λ) of LS, RA, and PW were 0.95, 0.94, and 1.04, respectively, and they were all close to 1 (Figure 4). This result, combined with QQ plots based on the observed and expected *p*-Values of the SNPs (Figure 4), indicated that there was negligible inflation caused by population stratification [50,51].

In addition to the population stratification, the cryptic relationship among individuals is another important reason for the inflation of false positives in GWAS, and considering the kinship of individuals in the GWAS model could decrease the influences [52,53]. An alternative way to derive kinship is relying on genetic markers, which more precisely specifies the actual difference between individuals than does relying on the pedigree, because some of these differences are not distinguishable when using the kinship derived from pedigree [54]. The best kinship to define the individual genetic relationship on a complex trait is the one derived from all the quantitative trait nucleotides (QTNs) underlying the trait [55], but the markers defining the kinship are always confounded with the tested markers and consequently decrease the statistical power of GWAS [30]. The SUPER method used in FarmCPU could dramatically reduce the number of genetic markers used to define individual relationships and remarkably decrease the confounding created by tested markers, because only the associated genetic markers are used to predict pseudo-QTNs [30]. In general, the number of pseudo-QTNs used as covariates for traits from 20 to 40 could well improve statistical power compared to deriving the overall kinship from all, or a random sample of genetic markers [30]. The statistical power could be doubled when using 26 pseudo-QTNs as covariates in the GWAS analysis of maize inbred lines [30]. In a simulation GWAS study of human data, 15–25 pseudo-QTNs were selected to define individual relationships, and the genetic background of individuals was well controlled in FarmCPU [29]. Another study also reported that using 40 QTNs as covariates in the GWAS study of wheat could reduce the false-positive rate [56]. In this study, the pseudo-QTNs used as covariates in the GWAS models were 33, 34, and 36, respectively (Figure 5). Although the selection of the pseudo-QTNs might not be able to capture all the effect caused by gene background, most of the effect could be captured by the SUPER method used in the random-effect model in FarmCPU [30]. The QQ plots and the inflation factors of the three traits all presented well (Figure 5), and we thought the effect of the cryptic relationship among individuals was effectively controlled in this study.

To gain insight into the function of the significant SNPs, genes that were in linkage disequilibrium regions (LD > 0.35) with these significant SNPs were used for further analysis. Of the 11 genes closest to the significant SNPs, some had been confirmed to be related to bone and muscle-tissue growth. *CDH12* was identified in chickens as a candidate core gene that could control the metatarsus circumference and regulate the traits of chest width and body weight [57]; the unusual expression of *PCDH9* in cells could lead to growth delay and microcephaly in humans [58]; *SH3BP4* could regulate the growth-factor-regulated mTORC1 pathway, which in turn had an impact on cell growth [59]; *USP6NL* was primarily implicated in epidermal growth factor in humans [60]; *CNTN3* was identified as a candidate gene that relates to the growth of corneal endothelial cells in the New Zealand rabbit [61]. Interestingly, we found that some of the 11 genes were also confirmed to be related to the reproductive performance of animals. For example, *CDH12* could regulate the development of the testis of chicken [57]; *TARP* participated in the process of AMPA receptor (AMPA-R) transporting, and affected the success rate of mouse reproduction and litter size [62,63]; *FSTL4* played a role in the expression of follicle-stimulating hormone (FSH), which in turn affected the ovarian follicular and corpus luteum dynamics, reproductive-hormone secretion, and estrus behavior of dairy cows [62,64]; the expression of *ATG4C* in endometrium was closely related to pregnancy status and affected the reproductive efficiency of beef heifers [65]. We suspect that although the three traits studied in this article were body-type traits, they were related to the reproductive performance of dairy cows (Figure 1). Therefore, some of the 11 candidate genes discovered in this article might affect the reproductive performance of animals by causing minor changes in body shape.

In the present study, a total of 45 genes within 200 Kb upstream and downstream of the significant SNPs of the three traits were found (Appendix A). The candidate genes of LS were mainly involved in the cell-adhesion progress, cell–cell adhesion via plasma-membrane adhesion molecule progress, and cell–cell junction organization progress, and some of them were related to muscle growth and development in cattle (Figure 7a). It has been reported that the cell-adhesion progress participated in the cell growth, muscle development, lipid metabolism, and fat deposition of beef cattle’s muscle [66]. The cell adhesion progress could also regulate the growth of bone cell [67], and could affect the growth performance of Ashidan yaks [68]. The cell–cell junction organization progress was a key factor that could affect the growth of the longissimus dorsi of beef cattle [69]. Just one KEGG pathway, named endocytosis, was enriched by the candidate genes of LS (Table 3), and the endocytosis pathway has been reported to participate in the muscle growth of Nellore cattle [70]. The *ARAP2* in the endocytosis pathway was a candidate gene that could affect the carcass traits, including carcass weight, eye-muscle area, back-fat thickness, and marbling of Korean cattle [71], and we surmised that *ARAP2* might be a candidate gene that could affect the LS of cows.

The candidate genes of RA were mainly involved in the protein-deglycosylation process and protein-modification process (Figure 7b). Studies have shown that the protein-deglycosylation process could affect osteogenesis and bone remodeling, and was critical for promoting bone morphogenetic protein signaling, which in turn affects bone morphology [72,73]. The protein-modification process has also been reported to participate in the bone morphogenesis of fetal bovine [74]. Three pathways, namely other glycan degradation, ECM–receptor interaction, and autophagy—other, were enriched by the candidate genes of RA (Table 3). It was reported that the ECM-receptor interaction pathway could participate in the osteoblast differentiation of fish [75], and autophagy was an important pathway affecting the growth and development of animal bones [76]. It is worth noting that the two genes in the ECM-receptor interaction pathway, *DMP1* and *DSPP*, were important genes that could regulate bone development and growth [77,78,79], and they might be candidate key genes that control the RA of dairy cows.

Two KEGG pathways; namely, the mRNA surveillance pathway and the RNA transport pathway, were enriched by the candidate genes of PW (Table 3), and the RNA transport pathway has been reported as potentially affecting the growth of rat bone cells [80]. It was noted that the *UPF2* gene participated in both pathways simultaneously, but there are few studies on the effect of *UPF2* on animal growth and development. *ECHDC3* is one of the candidate genes for PW (Appendix A), and *ECHDC3* is a kind of testis-tissue sperm-binding protein-encoded gene, and studies have shown that it was mainly related to metabolic disease and insulin sensitivity, and might affect the growth of animals [81,82]. We speculate that *UPF2* and *ECHDC3* might be key candidate genes that affect the PW of dairy cows.

## 5. Conclusions

This study focuses on the three body-shape traits, LS, RA, and PW, which were selected as indicators of the reproductive ability of Chinese Holstein cows. We estimated the genetic parameters of these three traits and performed genome-wide association analyses on them. The heritability of these three traits was of medium size, and a total of 11 significant SNPs were associated with them. We also found some candidate genes associated with LS, RA, and PW, and these genes might also play roles in these three traits of dairy cows. Bioinformatics analyses of candidate genes were also performed, and the pathways and the biological processes they enriched were presented. In general, our study found that the three body traits, LS, RA and PW, were closely related to the reproductive performance of dairy cows, and we detected some new variants and candidate genes that might affect these traits from a genetic perspective. Our findings provide useful biological information for the improvement of body shape traits and reproductive performance, and therefore will contribute to the genomic selection of Chinese Holstein cows.

## Figures and Tables

**Figure 1 animals-11-01927-f001:**
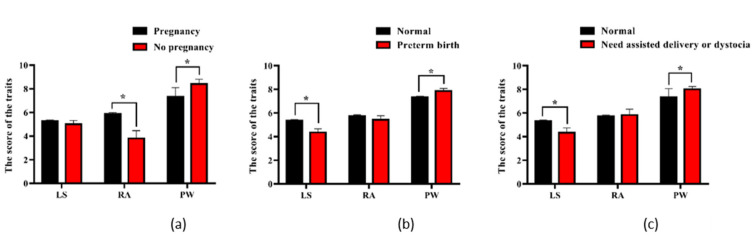
Relationship between the scores of Loin Strength (LS), Rump Angle (RA), and Pin Width (PW), and the reproductive performance, including the occurrence of pregnancy after one breeding (**a**), the occurrence of premature birth (**b**), and the ease of calving (**c**) (mean ± Standard error; the asterisks signify *p* < 0.05). The Loin Strength (LS), Rump Angle (RA), and Pin Width (PW) scores were related to the reproductive performance of cows.

**Figure 2 animals-11-01927-f002:**
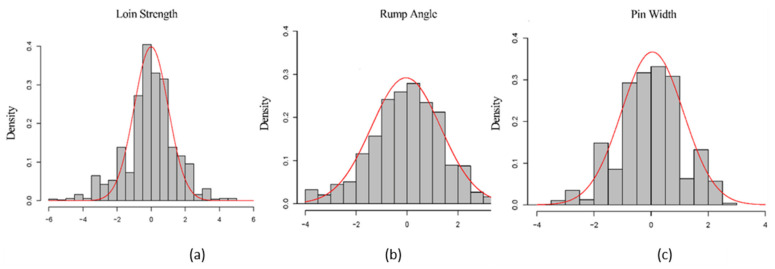
Frequency distribution of the adjusted phenotypes of Loin Strength (**a**), Rump Angle (**b**), and Pin Width (PW) (**c**), of the population in this study. The adjusted phenotypes of the three traits all presented approximately normal distributions.

**Figure 3 animals-11-01927-f003:**
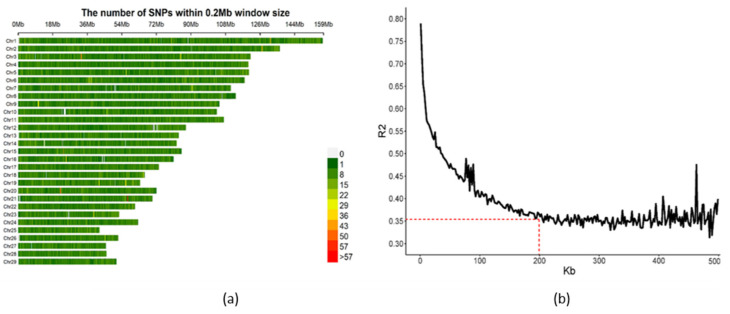
Genotyping chip information used in this study. (**a**) SNPs’ density distribution on 29 autosomes of the bovine genome. The SNP density was calculated per 0.2 Mbp window. (**b**) LD decay plot according to the average distance between SNPs. The R^2^ was lower than 0.35 when the average distance between SNPs was around 200 Kb.

**Figure 4 animals-11-01927-f004:**
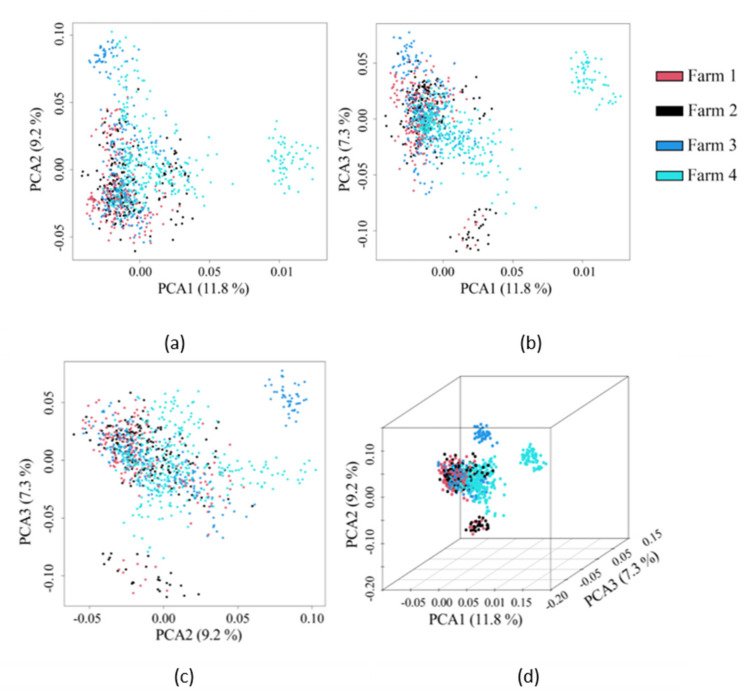
Population structure plots demonstrated by the 84,407 SNPs of 984 cows. The three highest principal components (PCs) were used to display the population structure by pairwise scatter plots (**a**–**c**) and the 3D plot (**d**). The PC1, PC2, and PC3 explained 11.8%, 9.2%, and 7.3% of the variation, respectively.

**Figure 5 animals-11-01927-f005:**
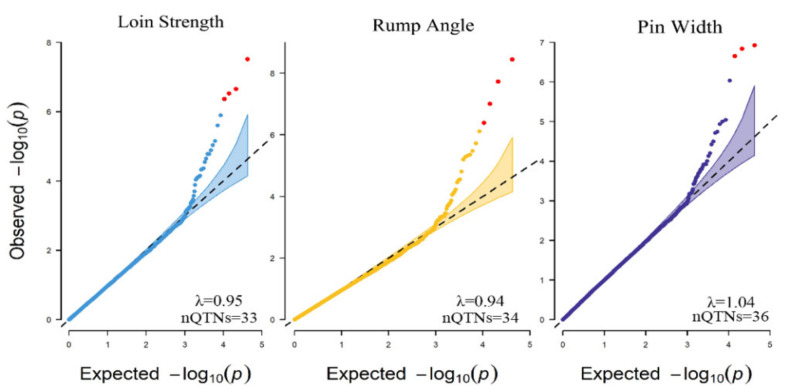
Quantile–quantile (QQ) plots of the three traits drawn by the expected *p*-Value (the uniformly distributed quantile from 0 to 1) and observed *p*-Value of each SNP. The red dots are SNPs that exceeded the threshold; the shaded parts are the confidence intervals. λ: genomic inflation factor; nQTNs: number of pseudo-QTNs in the traits.

**Figure 6 animals-11-01927-f006:**
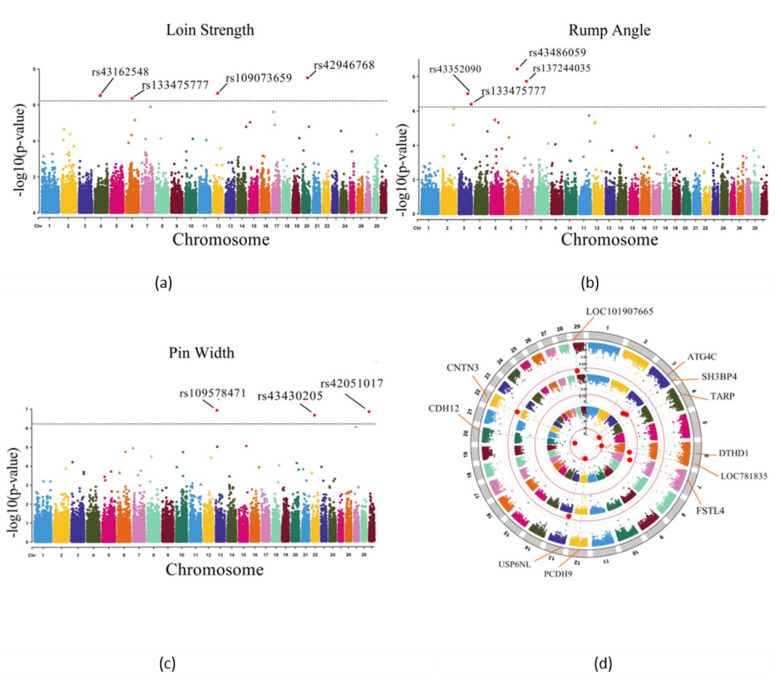
Manhattan plots of the LS (**a**), RA (**b**), and PW (**c**) drawn by the observed *p*-Value of each SNP and the location of the gene closest to each significant SNP of the three traits in the Manhattan plots (**d**). The gray horizontal lines in the Manhattan plots are significance thresholds (5.90 × 10^−7^); the red dots are SNPs that exceeded the threshold.

**Figure 7 animals-11-01927-f007:**
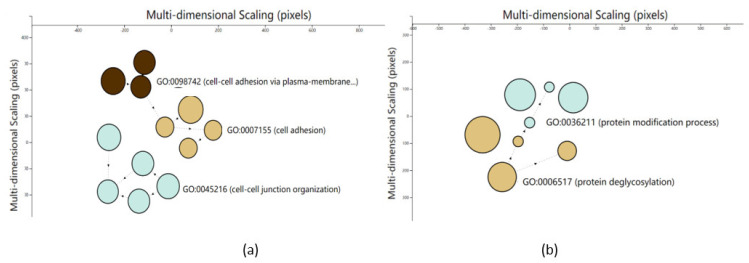
Clustering of enriched terms of LS and RA. In total, 12 GO terms (**a**) and 8 GO terms (**b**) were enriched in the biological process namespace using the genes within 200 Kb of the significant SNPs of LS and RA. (The results of PW are not listed because the genes within 200 Kb of the significant SNPs of PW were enriched to non-significant GO terms.) FunSet software automatically identified 3 and 2 clusters of LS and RA, respectively, using the eigengap approach [28].

**Table 1 animals-11-01927-t001:** Descriptive statistics for adjustment of LS, RA, and PW of cows; *n* = 984.

Traits	Arithmetic Mean	Minimum	Maximum	SD	CV (%)	Kurtosis	Skewness	h^2^ (SE)
LS	−0.08	−5.78	4.91	1.53	−18.19	3.84	−0.27	0.38 (0.05)
RA	−0.04	−4.95	3.58	1.37	−38.49	3.51	−0.35	0.22 (0.02)
PW	0.05	−5.45	2.56	1.09	23.67	3.56	−0.44	0.20 (0.02)

LS: Loin Strength; RA: Rump Angle; PW: Pin Width; SD: standard deviation; CV: coefficient of variation; h^2^: heritability; SE: standard error.

**Table 2 animals-11-01927-t002:** Information relating to the identified significant single-nucleotide polymorphisms (SNPs) and the nearest genes.

Trait	SNP	CHR	Position	Nearest Gene	Distance	MAF	Effect	EVG	*p*-Value
LS	rs42946768	20	51505605	*CDH12*	Within (intronic)	0.373476	−0.31	1.30%	3.08 × 10^−8^
	rs109073659	12	40319808	*PCDH9*	Within (intronic)	0.489837	0.29	1.22%	2.23 × 10^−7^
	rs43162548	4	50163217	*TARP*	5 Kb	0.172764	0.33	0.97%	2.99 × 10^−7^
	rs133475777	6	55719468	*DTHD1*	Within (intronic)	0.272358	0.29	0.91%	4.29 × 10^−7^
RA	rs43486059	6	102570596	*LOC781835*	Within (intronic)	0.489329	−0.29	1.38%	3.61 × 10^−9^
	rs137244035	7	45115020	*FSTL4*	Within (intronic)	0.455285	−0.28	1.32%	1.88 × 10^−8^
	rs43352090	3	82508654	*ATG4C*	200 kb	0.365854	−0.29	1.05%	9.91 × 10^−8^
	rs43366267	3	114684449	*SH3BP4*	50 Kb	0.318089	0.27	0.93%	4.10 × 10^−7^
PW	rs109578471	13	12679178	*USP6NL*	Within (intronic)	0.315041	−0.18	0.96%	1.18 × 10^−7^
	rs42051017	29	3370134	*LOC101907665*	200 Kb	0.21748	−0.20	0.87%	1.45 × 10^−7^
	rs43430205	22	26807183	*CNTN3*	200 Kb	0.272358	−0.18	0.87%	2.24 × 10^−7^

CHR: chromosome; LS: Loin Strength; RA: Rump Angle; PW: Pin Width; MAF: minor allele frequency; Effect: the regression coefficient of each variation; EVG: explained genetic variation.

**Table 3 animals-11-01927-t003:** Details of the pathways enriched by the genes within 200 Kb of the significant SNPs of traits.

Traits	Pathway	Description	Gene Name	*p*-Value
LS	bta04144	Endocytosis	*ARAP2*	0.0279
RA	bta00511	Other glycan degradation	*LOC781835, LOC523503*	0.0003
	bta04512	ECM-receptor interaction	*DSPP, DMP1*	0.0041
	bta04136	Autophagy—other	*ATG4C*	0.0361
PW	bta03015	mRNA surveillance pathway	*UPF2*	0.0113
	bta03013	RNA transport	*UPF2*	0.0210

LS: Loin Strength; RA: Rump Angle; PW: Pin Width.

## Data Availability

The data presented in this study are available on request from the corresponding author.

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
