# Peer review of "Genome-Wide Association Study on Reproduction-Related Body-Shape Traits of Chinese Holstein Cows"

_animals, 2021, doi:10.3390/ani11071927_

Round 1

Reviewer 1 Report

The authors in this work performed GWAS on three body shape traits that have previously been shown to affect reproductive traits, particularly calving traits. Fertility and reproduction are essential traits in animal farming, but despite this, only moderate progress has been made, partly because of the complexity of the traits. Therefore, finding genomic markers can promote genomic selection as well as an understanding of the biology and the genetics that affect fertility.

In general, the manuscript is well written. The methods and the experimental design are technically sound. However, I have several comments and points that I think will allow a better interpretation of this work's findings and their significance.

Methods.

Line 138. Why did the authors decide to remove the sex chromosome? doesn't the study included only cows? Please justify or include the sex chromosome SNPs in the analysis.

Results.

Line 185. The author should provide the correlation scores for other major traits, specifically reproductive and calving traits. I believe this data should be available and is essential for understanding the context of the tested conformational traits.

Line 205. I think it would benefit if the authors could provide the phenotype distribution across the farms.

Figure 4. how was the expected p-value calculate? 

Table 2. may the authors provide the  variation explained by each of the SNPs?

Table2. It seems the PW SNPs all fail to reach adj. significance of 5.9 × 10-7. Please clarify.

Line 252. why did the authors use all 45 genes from the luci of all the three different traits QTLs in the enrichment analysis? If I understood, there was no correlation, and there is no evidence that the traits are dependent. The authors should perform the enrichment analysis for each trait gene list. Otherwise, please justify this analysis.

Line 259. The results of this analysis (PPI) are not clear to me. Do the 13 proteins interact with each other or with other proteins outside the 45 tested proteins? Please clarify and explain the significance of this finding.

Discussion.

Line 333. the phenotypes studied in this work are body shape/conformation traits that might affect reproduction. Since the authors did not provide correlations scores for reproduction and calving traits, we cannot understand the extent to which these traits are associated. Thus the general association between the genes spanning the QTLs to reproduction and fertility is no supporting evidence. The authors should focus on body shape/ morphological/conformational/anatomical traits, as given in line 370.

The same goes for the enrichment analysis, line 348. Again, the authors studied body shape traits that might affect calving but not (directly) fertility and reproduction.

Minor. Line 183. did the authors meant "and PW"?

Author Response

Dear reviewer:

    We are very grateful to you for your constructive comments on our manuscript, the attached file is the response to your comment, thanks again.

Kinds regard

Authors of the manuscript

Reviewer 2 Report

The authors claim that Loin Strength (LS), Rump Angle (RA), and Pin Width (PW) are closely related to the reproductive performance of dairy cows and conducted a genome-wide association study for these three indicator traits.  I am not at all convinced that LS, RA, and PW are closely related to reproductive performance.  Why not perform a GWAS for reproductive traits themselves rather than for linear measurements that might serve as possible indicator traits?  The manuscript is poorly written and requires a great deal of editing.  The authors should work with a professional editing company to rewrite the paper.

lines 13-14:  "Cows with good reproductive performance could effectively reduce the occurrence of reproductive diseases".  It seems to me that the cause and effect are reversed here.  It is not that good reproductive performance would reduce the occurrence of reproductive diseases, but rather a reduction in reproductive diseases would enhance reproductive performance.

line 15:  "enhance" in place of "enhancing".

line 31:  "close" in place of "closed".

line 38:  Delete "work".

lines 43-44:  Change to "...important feature for livestock.  Successful pregnancy and...".

line 47:  Change to "multi-genes".

line 48:  Delete "sometimes".

line 53:  Change to "...indispensable activity in dairy cattle...".

lines 57-58:  Change to "in which a nine-point scoring system is used to...".

lines 58-60:  Please provide references for the following statement:  "Three traits among them are closely related to the reproductive performance of dairy cows, namely Loin Strength (LS), Rump Angle (RA), and Pin Width (PW)".

line 61:  I have not heard the term "waist" used in cattle.  Is there a more common term that could be used?

line 67:  "and" in place of "And".

line 76:  "ketosis" does not need to be capitalized.

line 79:  "the" in place of "The".  Also, change "regions" to "region".

line 80:  Change to "...was screened and found to be associated with...".

line 83:  Change to "Europe [17,18], there are few GWAS...".

lines 88-89:  Change to "...for improving the genomic selection of reproductive performance in dairy cows".

lines 91-92:  Change to

2.1. Ethics statement

Hair follicle sample collection and...:.

line 95:  Delete "strictly".

line 103 and elsewhere:  I do not think it is correct to refer to these three linear measurements as "reproductive traits".  They may be indicators of reproductive ability, but they are not "reproductive traits".

lines 103-108:  "three" in place of "3".

lines 107-108:  Change to "...were between 1 and 4 and the...".

line 109:  Change to "Adjustment of phenotypes for analysis".

line 117:  Change to "...normal distributions ? ~ ? (0,???2) and...".

line 124:  Delete "the followed".

line 152:  Change to "...GWAS models. The following is...".

line 158:  Change to "...vector of random residuals...".

line 161:  "...the same as in formula (2)...".

line 168:  Change to "...downstream of the significant...".

line 173:  Change "were" to "was".

line 183:  "was" in place of "were".

Table 1:  What is PS?  It is not defined in the footnote.  Why are means and minimum values negative?  I thought these three traits were recorded using a scale of 1 to 9.

lines 194-199:  Change to "After quality control, 84407 SNPs on 29 chromosomes remained for subsequent marker analysis. The distribution of the SNP information within 200 Kb windows on the different chromosomes is shown in Figure 2a. The change of LD decay with the increase of the average distance between SNPs in the current population is presented in Figure 2b.  The R2 was lower than 0.35 when the average distance between SNPs was around 200 Kb (Figure 2b)".

line 203:  Change to "...between SNPs was around 200 Kb".

line 205:  Delete "of PCA results".

line 206:  Delete "that".

line 214:  "84,407 SNPs".  This is the only place in the manuscript where you used a comma in 84407.

line 226:  Change to "...located on chromosomes 4, 6, 12, and 20, respectively, were detected...".

line 227:  Change to "...and the genes nearest to the four SNPs were...".

lines 229-230:  Change to "...located on chromosomes 3, 3, 6, and 7, respectively, were associated with RA, and the genes nearest to the four SNPs were...".

lines 232-233:  Change to "...located on chromosomes 12, 22, and 29, respectively, were associated with PW, and the genes nearest to the three SNPs...".

line 265:  The highlighted what?

lines 279-280:  Change to "Pablo estimated the heritability of the calving interval of Japanese Black cows ranging from 0.12 to 0.20...".

line 282:  Delete "and the heritability was about medium to low too".

lines 284-285:  Change to "...of environmental factors, such as age, climate, and nutritional status on the measurements [1]".

lines 285-288:  Change to "Although phenotypic correlations existed among LS, RA, and PW, the genetic correlations were low (< 0.1, Table S1).  Therefore, these three traits should be selected separately in dairy cattle".

lines 290-291:  The 2 in r2 should be a superscript.

lines 292-293:  Kb or kb?  Be consistent here and throughout the manuscript.

line 293:  Spell out the number to begin the sentence.

line 299:  Leave a space in "200kb".

line 301:  Be consistent in the use of spacing in "Chr7, Chr10, Chr 12, and Chr 16".  Carefully proofread the entire manuscript to eliminate the inconsistencies in spacing, capitalization, etc.

lines 309-311:  Change to "The stratification may have been caused by the semen used on the four farms coming from different countries, plus some of the semen may have been from local bulls".

lines 314 and 382:  "close" in place of "closed".

lines 314-316:  Change to "...(Figure 4).  This result combined with Q-Q plots based on the observed and expected p-values of the SNPs (Figure 4) indicated that the population stratification was successfully corrected [36,37]".

line 317:  Delete "detected".

line 323:  What are "micro-effect genes"?

line 330:  What is "it"?

line 331:  Delete "acquired".

lines 333 and 354:  Change to "affected".

line 337:  By "litter number" do you mean "litter size"?

line 358:  Change to "pathways".

line 363:  Delete "occurs".

line 365:  Do not begin a sentence with "And".

line 367:  Do not use contractions in formal writing.  Use "it is" rather than "it's".

line 369:  Change to "...protein network. The other..."

line 373:  "studies" in place of "studied".

line 381:  Delete "detected".

Author Response

(The authors gave the same response as above.)

Reviewer 3 Report

The paper argues that the three morphological traits are closely related to reproductive performance to justify that a GWAS analysis on these traits are of interest. No correlations are provided in the paper between these morphological traits and any reproductive trait, thus making the study doubtfully useful for the cited purpose. In my opinion, this makes the paper unacceptable. In addition, as a major concern, I am not convinced the population structure is completely removed with only the first three principal components.

Detailed revision

L99-102 The name of the farms is irrelevant

L132-133 I guess 5% instead of 95% as it is cited that this is the frequency of the minor allele.

L148-150 Please describe what are the association markers and how they operate here.

L157 Are the pseudo QTNs those previously called association markers? The model used here does not seem to be widely known and this model needs clarification.

L159 The SUPER algorithm is unknown for me and I guess for most of the readers. Explain it briefly.

L164-165 The Bonferroni reference is missing

L179-180 It should be better to provide a statistical test of normality.

L181 In table 1 the name of the traits are wrong or confusing. Is PS in reality RA?

L182-183 "...for LS, PS and were ..."

L186 I do not have access to table S1. Is it an editorial concern?

L205-210 The number of principal components to fit depend on in what extent they explain the global variability. On the light of the total variability explained and looking at the Figure 3, it is very probable that there is still a lot of structure defined by the next principal components. I think that  there must be several more components to include in the model.

L224-230 In Figure 5 the horizontal line defining the threshold of significance is not visible, and it looks that some of the selected SNPs falls under the theoretical threshold. It is also clear that this is so in table 2 for example for PW in which it looks none of them are significant, but also there is other in RA trait.

L252 I do not have access to table S2. Is it an editorial concern?

L254 I do not have access to table S3. Is it an editorial concern?

L282 Disagree. They are surprisingly moderate

L315-316 Disagree. In my opinion the indication of the figures is that this correction would have to be done, but not that the stratification was successfully corrected.

L317 What is FDR?

L318 Table 2 does not prove that these traits are associated to reproductive performance. In fact no evidence nor reference has been provided the authors to prove that.

L317-327 This paragraph (and most of the text) assume that these three traits are measuring the same concept. Correlations  between them show that they are rather independent and there is no evidence also they are really correlated to reproductive traits.

L377-378 Again, authors are very insistent on this unjustified idea that unfortunately invalids completely the study. The analysis done was not on cattle reproduction but on body shape.

L376 This is not a results section. This is the place where the authors must summarize the utility of the analyses and their implications.

Author Response

(The authors gave the same response as above.)

Round 2

Reviewer 1 Report

The authors made efforts to address the comments of the reviewers. Overall, GWAS performed well, but my main concern remains regarding the relevance to reproductive traits. In the introduction, the authors cite several works that have linked conformational traits with reproduction. I suggest they elaborate on the association between the traits. What are the degree and the direction of correlation?

The authors added a statistical test to several reproduction traits. Yet, they did not specify how they measured the traits and whether they made a statistical adjustment, such as parity, farm, year of birth, AI technician, season, etc. If (at all) the correlation between the conformational traits and reproductive traits is small. And the effect of the genetic markers on the conformational traits is minor; then, I wonder what the significance of the finding to reproduction?

In figure 1: what are the asterisks (p < 0.05?) Should the authors correct for multiple testing (9 tests in this case)?

Table 2: please provide the regression coefficient and the effect size of the adjusted phenotype to each variation

Table 3: If I understood correctly, in some of the detailed pathways, the authors found only one gene? If so, I wonder what does one gene means in enrichment analysis?

Reviewer 2 Report

Thank you for addressing the comments and concerns from my first review of this paper.  Please also make the following changes.

line 17:  Change "calculated" to "estimated".  We never know the true values of the genetic parameters, and, therefore, we must estimate rather than calculate them.

line 44:  Change to "...pregnancy and birth of offspring can maintain...".  "Childbirth" refers to the birth of human babies.

line 204:  As I pointed in my first review of this paper, "waist" is not a term that is usually applied to cattle.  Can you use a better term?

line 233:  Change to "The collection of hair follicle samples and the measurement of traits...".

lines 340-341:  Did you use the average of the measurements taken by the different technicians?

line 371:  Suggest changing to "reference genome".

line 376:  Change to "genotypes".

line 519:  Change to "...is a vector of random residuals...".

line 523:  "chose" in place of "chosen".

line 524:  Change to "...selected the most influential bins...".

lines 632-633:  I do not understand what is meant by "regression coefficient of adjusted phenotype to each variation".  Perhaps this could be reworded to improve the clarity of the statement.

line 635:  Change to "...correction method [30].  The type I error rate was...".

line 641:  Change to "reference genome".

line 652:  Change to "...the RA and PW scores significantly affected the...".

line 654:  Change "correlated" to "associated".  You did not calculate correlations; you tested for differences in means.

lines 652-660:  Two asterisks (as shown in the figure) normally indicate highly significant differences in means (P < 0.01).

Figure 2(c):  Add a label to the Y-axis.

line 749:  The 2 should be a superscript in h2.

Table 2:  Define EVG in the footnote.

line 898:  Change to "...genes of each trait were significantly enriched in the...".

line 102?:  Change "representatively" to "respectively".  Delete "by".

line 1035-1036:  Change to "Due to the limited management in some small...".

lines 1044-1045:  Change to "...pregnancy after one breeding in the first parity of cows.  A higher LS score could significantly reduce the...".

lines 1046-1047:  "...the three traits used in this study were reasonable...".

line 1048:  Delete "in breeding".

lines 1049-1050:  The following sentence is unclear and needs to be reworded:  "Therefore, attempts to improve the body traits of dairy cows should be adjusted in accordance with the actual conditions of dairy cows and pastures, and should not only blindly pursue trait scores".

lines 1202-1203:  Change to "The level of LD (r2) between SNPs decreased as the distance increased...".

line 1205:  Change to "...used to search for candidate genes that...".

line 1207:  Change to "...for genes in other GWAS...".

lines 1208-1210:  Change to "...was much lower than in Simmental cattle, Wagyu cattle, and Iranian Water Buffalo, which indicated that the degree of artificial selection of Chinese Holstein cows was higher than in beef cattle...".

line 1217:  Change to "...genetic structures are included in a GWAS study, the genetic...".

line 1219:  Change "resulted" to "result".

line 1241:  Change to "...reproductive performance of animals.  For example, CDH12 could regulate...".

line 1494:  Change to "...animals by causing minor changes in body...".

line 1510:  Change to "...we surmised that ARAP2 might be a...".

line 1524:  Delete "in this study".

line 1533:  Change to "...might be key candidate key genes that affect...".

line 1537:  Change "calculated" to "estimated".

lines 1890-1893:  I suggest using a stronger concluding statement such as, "Our findings provide useful biological information for the improvement of body shape traits and reproductive performance, and, therefore, will contribute to the genomic selection of Chinese Holstein cows".

line 1894:  The following are available online at www.mdpi.com/xxx/s1, Figure S1.  When I click on this link, I get a message that says File Not Found.
